# Online Lazy Gradient Descent is Universal on Strongly Convex Domains

**Daron Anderson**
School of Computer Science and Statistics
Trinity College Dublin

**Douglas Leith**
School of Computer Science and Statistics
Trinity College Dublin

## Abstract

We study Online Lazy Gradient Descent for optimisation on a strongly convex domain. The algorithm is known to achieve $O(\sqrt{N})$ regret against adversarial opponents; here we show it is universal in the sense that it also achieves $O(\log N)$ expected regret against i.i.d opponents. This improves upon the more complex meta-algorithm of Huang et al [20] that only gets $O(\sqrt{N \log N})$ and $O(\log N)$ bounds. In addition we show that, unlike for the simplex, order bounds for pseudo-regret and expected regret are equivalent for strongly convex domains.

## 1 Introduction

Online linear optimisation is a repeated game with one opponent. On turn $n$ we know the *cost vectors* $a_1, a_2, \ldots, a_{n-1} \in \mathbb{R}^d$ and select an *action* $x_n$ from the *domain* $X \subset \mathbb{R}^d$. The opponent observes $x_n$ and chooses the next cost vector $a_n \in \mathbb{R}^d$ and we pay cost $a_n \cdot x_n$. Our goal is to select actions to make the *regret* $\sum_{i=1}^N a_i \cdot (x_i - x^*)$ small for $x^* \in \operatorname{argmin}\{\sum_{i=1}^N a_i \cdot x : x \in X\}$.

If the cost vectors are selected by an adversarial opponent then $a_1, a_2, \ldots, a_{n-1}$ give no information about $a_n$ and selecting a good action seems like an impossible task. Despite this, there are algorithms that get $O(\sqrt{N})$ regret against such opponents. The loss in performance against $x^*$ is $O(1/\sqrt{N})$ per turn which vanishes as $N \to \infty$. The central algorithms in the field are Hedge [18, 34]; Prod [9]; and (sub)Gradient Descent, which has Greedy [41] and Lazy [34] variants. All three algorithms get $O(\sqrt{N})$ regret. The first two are specialised to $X$ the simplex. The latter works for any compact convex domain. The bulk of online optimisation literature is refinements of the central algorithms.

In many cases however the cost vectors are non adversarial. For example they might be determined by fluctuations in traffic, the weather or the stock market over a short period of time. The standard model of *easy* opponents is cost vectors drawn independently from some fixed distribution. In the easy setting the history $a_1, \ldots, a_n$ provides information about the next cost vector. Indeed $\widetilde{a}_{n+1} = \frac{a_1 + \ldots + a_n}{n}$ is an increasingly reliable sequence of estimates for $a = \mathbb{E}[a_{n+1}]$ and it seems reasonable to play $x_{n+1}$ to minimise $\widetilde{a}_{n+1} \cdot x$. This is called Follow-the-Leader (FTL). For example if $X$ is the simplex FTL gives pseudo-regret $\mathbb{E}\left[\sum_{i=1}^N a \cdot (x_i - y^*)\right] \leq O(1)$ for $y^* \in \operatorname{argmin}\{a \cdot x : x \in X\}$. Note this is weaker than bounding the expected regret $\mathbb{E}\left[\sum_{i=1}^N a_i \cdot (x_i - x^*)\right]$. On the other hand FTL gives no guarantee against adversarial opponents, and we might not know in advance whether the opponent is i.i.d or adversarial. Hence we would like an algorithm that is *universal* in that it always gets $O(\sqrt{N})$ regret but specialises to a much stronger bound against i.i.d opponents.

Recently Huang et al [20] solved this problem for smooth, strongly convex domains. For these domains they showed FTL gives $O(\log N)$ regret if $\|\sum_{i=1}^n a_i\|$ grows linearly.[1] The result was generalised to nonsmooth domains by [22, 26]. Huang et al then combine FTL with the $(A,B)$-Prod meta-algorithm of [31] to ensure $O(\sqrt{N \log N})$ regret against adversarial opponents. In this work

---

[1] Without smoothness they show the growth condition occurs with high probability against i.i.d opponents.

35th Conference on Neural Information Processing Systems (NeurIPS 2021).

---

**Algorithm 1:** Online Lazy Gradient Descent

---

**Data:** Cost vectors $a_1, a_2, \ldots \in \mathbb{R}^d$. Parameter $\eta > 0$. Domain $X \subset \mathbb{R}^d$.

1 **for** $n = 0, 1, \ldots$ **do**

2     $x_{n+1} = \Pi_X \left( -\frac{\eta}{\sqrt{n}} \sum_{i=1}^n a_i \right)$

3     Receive $a_{n+1}$ and pay cost $a_{n+1} \cdot x_{n+1}$

---

we improve and extend their results. In particular we show the meta-algorithm is unnecessary as the much simpler Online Lazy Gradient Descent already achieves $O(\sqrt{N})$ and $O(\log N)$ bounds.

## Terminology and Notation

Throughout $\| \cdot \|$ is the Euclidean norm. Given a vector subspace $V \subset \mathbb{R}^d$ and linear operator $A : V \to V$ write the operator norm as $\|A\| = \max\{\|Av\| : v \in V, \|v\| = 1\}$. It is known (see [10] Theorems 4.1 and 4.2) that $\|A\| = \max_j \sqrt{|\lambda_j|}$ for $\lambda_j$ the eigenvalues of $A^T A$. Moreover if $A$ is symmetric we have $\|A\| = \max_j |\mu_j|$ for $\mu_j$ the eigenvalues of $A$. The set $Z \subset \mathbb{R}^d$ is called $\lambda$-strongly convex to mean for each $x, y \in Z$ and $\alpha \in [0, 1]$ the ball $B(\alpha x + (1-\alpha)y, r)$ is contained in $Z$ for $r = \frac{\lambda}{2}\alpha(1 - \alpha)\|x - y\|^2$. The pseudo-regret $\mathbb{P}[R_N]$ and expected regret $\mathbb{E}[R_N]$ are defined:

$$\mathbb{P}[R_N] = \mathbb{E}\left[\sum_{i=1}^N a \cdot x_i - \min_{x \in X} \sum_{i=1}^N a \cdot x\right] \qquad \mathbb{E}[R_N] = \mathbb{E}\left[\sum_{i=1}^N a_i \cdot x_i - \min_{x \in X} \sum_{i=1}^N a_i \cdot x\right] \quad (1)$$

The Jensen Inequality implies $\mathbb{P}[R_N] \leq \mathbb{E}[R_N]$. We write $\Pi_X$ for the Euclidean projection onto the domain $X$. The Online Lazy Subgradient algorithm is given as Algorithm 1. Note this is different from the so-called Greedy Subgradient algorithm with starting point $x_1 = 0$ and recursive update $x_{n+1} = \Pi_X(x_n - \eta a_n / \sqrt{n})$. Section 5 Figure 2(b) suggests Greedy Subgradient is not universal.

## Summary of Contributions

Throughout we make the following assumptions on the domain and cost vectors. Both assumptions are nontrivial. For example Assumption 1 holds on the Euclidean and $\ell_p$ balls for $p \in (1, 2]$ but not for the simplex or polytopes. Assumption 2 on the cost vectors is a typical model of an *easy opponent* and fails if the opponent is intelligent and adapts to our past moves.

**Assumption 1.** The domain $X \subset \mathbb{R}^d$ is compact, $m$-strongly convex and contains the origin. Write $D = \max\{\|x\| : x, y \in X\}$. The boundary $\mathcal{M} = \partial X$ is a $(d-1)$-dimensional $C^2$ manifold. Namely each $z \in \mathcal{M}$ has a neighborhood $U$ in $\mathbb{R}^d$ and $C^2$ function $F : U \to \mathbb{R}$ with nonzero gradient such that $\mathcal{M} \cap U = \{x \in U : F(x) = 0\}$. Such a function is called a **coordinate patch** at $z$.

Henceforth write $N(x)$ for the outwards unit normal at $x \in \mathcal{M}$. To represent $N : \mathcal{M} \to \mathbb{R}^d$ locally we can choose a coordinate patch $F : U \to \mathbb{R}$ at $x^*$ and write $\nabla N = \frac{\nabla F}{\|\nabla F\|}$. Since $F$ is $C^2$ the matrix $\nabla N$ of partial derivatives exists.

**Assumption 2.** The cost vectors $a_1, a_2, \ldots$ are i.i.d with expectation $\mathbb{E}[a_n] = a$. For all $n$ we have $\|a_n\| \leq L$ and $\|a_n - a\| \leq R$ and $\mathbb{E}\|a_n - a\| \leq \delta$. Let $x^* \in \operatorname{argmin}\{a \cdot x : x \in X\}$ be the expected minimiser. The domain $X$ being strongly convex implies $x^*$ is unique.

Our main result is Theorem 3 which says Online Lazy Gradient Descent achieves $O\left(\frac{L^2}{m\|a\|}\log N\right)$ expected-regret under Assumptions 1 and 2. This bound is essentially tight by [20] Theorem 9. The improved behaviour against i.i.d opponents is remarkable since the algorithm was designed with the adversarial case in mind, and predates the recent interest in universal algorithms. Our second contribution is Theorem 2 which states the gap between expected regret and pseudo-regret is $O\left(\frac{L^2}{m\|a\|}\right)$. Theorem 2 is of independent interest since it fails if the domain is the simplex.

From a technical standpoint, our Gradient Descent analysis goes beyond that of [20, 22, 26] for FTL. Their central idea is that the normal vector to $\mathcal{M}$ at $x_n \in \operatorname{argmin}\{(a_1 + \ldots + a_{n-1}) \cdot x : x \in X\}$ points along $A_{n-1} = -\frac{a_1 + \ldots + a_{n-1}}{n}$. Using $\|A_n - A_{n-1}\| = O(1/n)$ and how strong convexity says the unit normal changes quickly as we vary the basepoint, they prove $\|\Theta_n - \Theta_{n+1}\| = O(1/n)$ for the normal vectors $\Theta_n, \Theta_{n+1}$ at $x_n, x_{n+1}$. Summing the resulting series gives a logarithmic regret

bound. Unfortunately this method is insufficient to analyse Gradient Descent instead of FTL as it only gives $\|\Theta_n - \Theta_{n+1}\| = O(1/\sqrt{n})$. and the resulting series give only an $O(\sqrt{N})$ bound. Thus we follow a new approach where we prove a high probability bound $\|N(x_n) - N(x^*)\| = O(1/\sqrt{n})$ for the unit normals. For comparison existing works do not mention the expected minimiser $x^*$ at all. Then we use strong convexity to see the boundary surface locally looks like a quadratic with axis pointing along $-a$. Hence $\|x_n - x^*\| = O(1/\sqrt{n})$ and only $O(1/n)$ of the displacement is in the $a$-direction; and the larger orthogonal component does not contribute to pseudo-regret. Summing the series gives an $O(\log N)$ bound for pseudo-regret. To convert this into a bound for expected regret we use our Theorem 2 which says the gap between expected regret and pseudo-regret is finite.

Our differential geometry methods are more nuanced than existing works. Existing works consider only the magnitude $\|x_n - x_{n+1}\|$ but we consider both the magnitude and direction of $x_n - x^*$. The relevant object is $\nabla N(z)v$ in Lemmas 1-5 and Proposition 1 which captures the rate of change of the unit normal as we perturb the basepoint in the $v$-direction.

## 2   Related Work

The first strand of related work is *universal algorithms*. That is to say online algorithms with $\widetilde{O}(\sqrt{N})$ regret against adversarial opponents that specialise to get stronger bounds against i.i.d opponents. The original research focus was on the simplex and bandit feedback [4, 8, 32, 33, 37, 40]. In the full information setting the FlipFlop algorithm [11] interpolates between FTL and Hedge on the simplex to get $O(1)$ and $O(\sqrt{N \log N})$ regret against i.i.d and adversarial opponents respectively. The $(A,B)$-Prod meta-algorithm [31] does something similar in much greater generality. Another result comes from [16]. They design a variant of the Prod algorithm [9] with different learning rates for each arm. As a corollary they get $O(1)$ expected regret against i.i.d opponents with a unique best arm. Apart from [31] all the above algorithms apply only to the simplex.

However recent results suggest such complicated algorithms are unnecessary. In the full information setting it was recently proved that two of the central algorithms do much better against i.i.d opponents. Hedge gets $O(\log(d)L_\infty^2/\Delta)$ pseudo-regret on the simplex [27]; and Online Lazy Gradient Descent gets $O(L_2^2/\Delta)$ pseudo-regret on polytopes[2] [1, 2].The latter is notable since it works for domains other than the simplex. Here $d$ is the dimension; $\Delta$ the suboptimality gap; and we assume $\|a_n\| \le L_2$ and $\|a_n\|_\infty \le L_\infty$ where appropriate. The above bounds are in addition to the $O(\sqrt{N})$ regret bounds of Gradient Descent and Hedge (see [34] Chapter 2) against adversarial opponents.

The second strand of related work is the study of online algorithms on strongly convex domains. Typically these are unit balls with respect to some norm. For example [39] consider the $\ell_p$ unit ball with $p \in (1, 2]$ in the full-information setting. Another setting of interest [5, 12, 19] is when predictions are available for the direction of the next cost vector. In the bandit setting [7] consider $\ell_p$ balls; [6] consider the Euclidean ball; and [30] consider general strongly convex sets. They allow infinitely many arms but require the cost vectors be stochastically generated. In the offline setting [17] show the Frank-Wolfe method has $O(1/n^2)$ convergence rates − and hence finite regret − when the domain is strongly convex. [23] prove similar results for more general so-called uniformly convex sets. Under additional conditions the older works [13, 14, 25] show an exponential convergence rate.

## 3   Bound for Pseudo-Regret

Here we show Online Lazy Gradient Descent gets $O(\log N)$ pseudo-regret against i.i.d cost vectors on a $C^2$ strongly convex domain. This is in addition the $O(\sqrt{N})$ regret bound for adversarial costs.

**Theorem 1.**  Under Assumptions 1 and 2, Algorithm 1 with parameter $\eta > 0$ achieves pseudo-regret

$$\mathbb{P}[R_N] \le \frac{144}{m\|a\|}\left(\frac{\sqrt{8\pi}DR}{\eta} + 4R^2\right) \log N + O(1).$$

In particular for hyperparameter $\eta = \Theta(D/L)$ we have $\mathbb{P}[R_N] \le O\left(\frac{L^2}{m\|a\|} \log N\right)$.

By [34] Theorem 2.4 hyperparameter $\eta = \Theta(D/L)$ is optimal for both adversarial and i.i.d bounds.

---

[2]Huang et al [20] already proved an $O(1)$ bound for polytopes with unique optimal vertex. However dependence on dimension and suboptimality gap is not obvious.

We begin with some geometric intuition for our proof strategy. To interpret Lemma 1 it is convenient to imagine the domain is translated to put $z$ at the origin and the outward unit normal $N(z)$ points along the $d$-th coordinate direction (henceforth called the height). Since $N(z)$ has unit length, the left-hand-side of Lemma 1 is the height of the boundary at $x$. The lemma says the height changes quadratically with coefficient depending on the direction $v$ as defined below.[3]

**Definition 1.** Given points $x, y \in \mathcal{M}$ a **geodesic** from $x$ to $y$ is a $C^2$ path in $\mathcal{M}$ with endpoints $x$ and $y$ with $\|\gamma'(t)\| = 1$ for each $t$, and such that $\gamma$ has minimal length among such paths. That length is called the **geodesic distance** $d(x, y)$. The unit vector $v \in \mathbb{R}^d$ is called a **direction** from $x$ to $y$ to mean there is a geodesic $\gamma : [0, d(x, y)] \to \mathcal{M}$ from $x$ to $y$ with $\gamma'(0) = v$.

In Lemma 6 we plug the expected minimiser $z = x^*$ into Lemma 1. The height then points along $a$ and the left-hand-side of the bound becomes the pseudo-regret of $x$. We then (a) remove the dependence on direction and bound the coefficient by $1/m$ and (b) use strong convexity to convert the right-hand-side into the squared difference $\|N(x) - N(x^*)\|^2$.

Finally we plug $x = x_n$ into Lemma 6 and use Lemma 7 to see $\|N(x_n) - N(x^*)\|^2 = O(1/n)$ with high probability. The idea is to write the unprojected action $y_n = -\frac{\eta}{\sqrt{n}} \sum_{i=1}^n a_i$ as $y_n = -\eta\sqrt{n}a + \frac{\eta}{\sqrt{n}} \sum_{i=1}^n (a - a_i)$. The first term points along $-\frac{a}{\|a\|} = N(x^*)$ and has length $\Omega(\sqrt{n})$. The second term is $O(1)$ with high probability due to the i.i.d sum. Hence after normalisation the component of $y_n$ orthogonal to $-\frac{a}{\|a\|}$ has size $O(1/\sqrt{n})$. Since the domain is bounded the same holds if we instead normalise $y_n - w$ for any $w \in X$. In particular taking $w = x_n$ we get $\left\| \frac{y_n - x_n}{\|y_n - x_n\|} - \left( -\frac{a}{\|a\|} \right) \right\| = O(1/\sqrt{n})$. Since $x_n = \Pi_X(y_n)$ we have $\frac{y_n - x_n}{\|y_n - x_n\|} = N(x_n)$. Since $-\frac{a}{\|a\|} = N(x^*)$ we get $\|N(x_n) - N(x^*)\|^2 = O(1/n)$ with high probability. Sum the series to get $P_N = O\left( \sum_{n=1}^N \frac{1}{n} \right) = O(\log N)$.

**Lemma 1.** For each $z \in \mathcal{M}$ there exists a coordinate patch $F : U \to \mathbb{R}$ such that for each $x \in U \cap \mathcal{M}$ and direction $v$ from $z$ to $x$ we have

$$|N(z)(x - z)| \leq \frac{|v^T \nabla^2 F(z) v|}{\|\nabla F(z)\|} d(x, z)^2.$$

Lemma 1 is proved using Lemmas 2-5 and Proposition 1 which are in turn proved in Appendix II. The arguments there use only routine differential geometry, but we provide self-contained proofs, since the literature focuses on intrinsically defined surfaces, rather than surfaces embedded in $\mathbb{R}^d$.

**Lemma 2.** For each $z \in \mathcal{M}$ and unit vector $v$ tangent to $\mathcal{M}$ at $z$ we have $\|\nabla N(z)v\| \geq m$.

**Lemma 3.** For each coordinate patch $F$ at $z$ and unit tangent $v$ to $\mathcal{M}$ at $z$ we have $\frac{v^T \nabla^2 F(z) v}{\|\nabla F(z)\|} \geq m$.

Lemma 4 says the unit normal function has bounded second derivative, despite any complications caused by the function being only locally defined.

**Lemma 4.** There exist constants $M, r > 0$ such that each $z \in \mathcal{M}$ has a coordinate patch $F : B(z, r) \to \mathbb{R}$ with $\|\nabla N(y)\|, \frac{\|\nabla^2 F(y)\|}{\|\nabla F(y)\|} \leq M$ for all $y \in B(z, r)$.

**Lemma 5.** Let $M$ be the constant from Lemma 4. For each geodesic $\gamma : [0, d(x, y)] \to \mathcal{M}$ from $x$ to $y$ we have for $d = d(x, y)$ and all $t \leq d$ the inequalities

$$(1) \ \|\gamma''(t)\| \leq M \qquad (2) \ \|\gamma(t) - \gamma(0) - t\gamma'(0)\| \leq \frac{M}{2}t^2 \qquad (3) \ \left| \|y - x\| - d \right| \leq \frac{M}{2}d^2$$

**Proposition 1.** Each $z \in \mathcal{M}$ has a neighborhood $U$ in $\mathbb{R}^d$ such that for all $x \in U \cap \mathcal{M}$ and direction $v$ from $z$ to $x$ we have

$$\|N(x) - N(z)\| \geq \frac{\|\nabla N(z)v\|}{2} d(x, z)$$

*Proof of Lemma 1.* Let $F, M, r$ be from Lemma 4. Since $F$ is $C^2$ Taylor's theorem says there is a function $\psi(t)$ defined on some neighborhood of the origin with $\lim_{t \to 0} \psi(t) = 0$ and

$$F(x) = F(z) + \nabla F(z)(x - z) + \frac{(x - z)^T \nabla^2 F(z)(x - z)}{2} + \psi(z - x)\|z - x\|^2$$

---

[3]It is of course possible to define geodesics and directions using a non-Euclidean notion of distance. However in this paper we only consider Euclidean geodesics.

Hence there is $r > 0$ such that for all $x \in B(z, r)$ we have $|\psi(z - x)| \leq \frac{m}{4} \|\nabla F(z)\|$ and so

$$\left\| F(x) - F(z) - \nabla F(z)(x - z) - \frac{(x-z)^T \nabla^2 F(z)(x-z)}{2} \right\| \leq \frac{m}{4} \|\nabla F(z)\| \|x - z\|^2.$$

Shrink $r$ if necessary to ensure $B(z, r)$ has geodesic diameter at most $\min\{m/2M^2, \sqrt{2m}/M^{3/2}\}$. By assumption $F(x) = F(z) = 0$. Hence for $d = d(x, z)$ the above gives

$$\left\| \nabla F(z)(x - z) - \frac{(x-z)^T \nabla^2 F(z)(x-z)}{2} \right\| \leq \frac{m}{4} \|\nabla F(z)\| \|x - z\|^2 \leq \frac{m}{4} \|\nabla F(z)\| d^2.$$

Let $v$ be a direction from $z$ to $x$. For $w = (x - z) - dv$. The reverse triangle inequality gives

$$\left\| |\nabla F(z)(x - z)| - \frac{|(dv)^T \nabla^2 F(z)(dv)|}{2} \right\| \leq \frac{m}{4} \|\nabla F(z)\| d^2 + |w^T \nabla^2 F(z)(dv)| + \frac{|w^T \nabla^2 F(z)w|}{2}. \quad (2)$$

Lemma 5 says $\|w\| \leq Md^2/2$ and Lemma 4 says $\frac{\|\nabla^2 F(z)\|}{\|\nabla F(z)\|} \leq M$. Hence we get

$$|w^T \nabla^2 F(z)(dv)| \leq \|w\| \|\nabla^2 F(z)\| \|dv\| \leq \frac{M}{2} \frac{\|\nabla^2 F(z)\|}{\|\nabla F(z)\|} \|\nabla F(z)\| d^3$$
$$\leq \frac{M^2}{2} \|\nabla F(z)\| d^3 \leq \frac{\|\nabla F(z)\|}{4} m d^2.$$

where the last inequality uses the assumption $d \leq m/2M^2$. Similarly $d \leq \sqrt{2m}/M^{3/2}$ gives $|w^T \nabla^2 F(z)w| \leq \frac{\|\nabla F(z)\|}{2} m d^2$. Hence (2) becomes

$$\left\| |\nabla F(z)(x - z)| - \frac{|v^T \nabla^2 F(z)v|}{2} d^2 \right\| \leq \frac{m}{4} \|\nabla F(z)\| d^2 + \frac{m}{4} \|\nabla F(z)\| d^2 = \frac{m}{2} \|\nabla F(z)\| d^2.$$

and so $|\nabla F(z)(x - z)| \leq \left( \frac{|v^T \nabla^2 F(z)v|}{2} + \frac{m}{2} \|\nabla F(z)\| \right) d^2$. Since the unit normal is $N(z) = \frac{\nabla F(z)}{\|\nabla F(z)\|}$ we can divide both sides to get $N(z)(x - z)| \leq \frac{1}{2} \left( \frac{|v^T \nabla^2 F(z)v|}{\|\nabla F(z)\|} + m \right) d^2$. Lemma 3 says $m \leq \frac{|v^T \nabla^2 F(z)v|}{\|\nabla F(z)\|}$. This completes the proof. $\quad\square$

**Lemma 6.** The minimiser $x^*$ has a neighborhood $U$ in $\mathbb{R}^d$ such that each $x \in U \cap \mathcal{M}$ has

$$a \cdot (x - x^*) \leq 4 \frac{\|a\|}{m} \|N(x) - N(x^*)\|^2.$$

*Proof.* Let $F : U \to \mathbb{R}$ be a coordinate patch at $z = x^*$ that meets the conditions in Lemma 1 and Proposition 1. Since $x^* = \operatorname{argmin}\{a \cdot x : x \in X\}$ the outwards unit normal is $N(x^*) = -\frac{a}{\|a\|}$ and the lemma gives

$$|a \cdot (x - x^*)| \leq \|a\| \frac{|v^T \nabla^2 F(x^*)v|}{\|\nabla F(x^*)\|} d(x, x^*)^2.$$

For all $x \in U$ and direction $v$ from $x^*$ to $x$ the proposition gives $d(x, x^*) \leq 2 \frac{\|N(x) - N(x^*)\|}{\|\nabla N(x^*)v\|}$. Combine the two bounds to get

$$|a \cdot (x - x^*)| \leq 4 \|a\| \left( \frac{|v^T \nabla^2 F(x^*)v|}{\|\nabla F(x^*)\|} \frac{1}{\|\nabla N(x^*)v\|^2} \right) \|N(x) - N(x^*)\|^2.$$

To complete the proof Lemma 3 says $\frac{|v^T \nabla^2 F(x^*)v|}{\|\nabla F(x^*)\|} = |v^T \nabla N(x^*)v| \leq \|v\| \|\nabla N(x^*)v\| \leq \|N(x^*)v\|$. Hence the middle factor above is at most $\frac{1}{\|N(x^*)v\|}$ which is at most $1/m$ by Lemma 2. $\quad\square$

The proofs of Lemmas 7 and 8 in Appendix III follow easily from the vector martingale theorems of Pinelis [28]. The use of these theorems − stated only for bounded martingales − is the major obstruction to generalising our main results to unbounded but subgaussian cost vectors.

**Lemma 7.** For each $n \geq 16D^2/\eta^2 \|a\|^2$. The Online Lazy Gradient Descent actions $x_1, x_2...$ give

$$\mathbb{E}\|N(x^*) - N(x_{n+1})\|^2 \leq \frac{1}{n} \frac{36}{\|a\|^2} \left( \frac{\sqrt{8\pi} DR}{\eta} + 4R^2 \right) + 8 \exp\left( -\frac{\|a\|^2}{32R^2} n \right).$$

**Lemma 8.** For each neighborhood $U$ of $x^*$ in $\mathbb{R}^d$ the Online Lazy Gradient Descent actions $x_1, x_2, \ldots$ give $\sum_{i=1}^{\infty} P(x_i \notin U) < \infty$.

*Proof of Theorem 1.* It is enough to prove a tail bound $\mathbb{E}\big[\sum_{i>n_0}^{N} a \cdot (x_i - x^*)\big] \leq O(\log N)$ for some fixed $n_0$. To use Lemma 7 let $n_0 = \lfloor 16D^2/\eta^2 \|a\|^2 \rfloor$.[4] Let $U$ be a neighborhood of $x^*$ that meets the conditions of Lemma 6. Since each $a \cdot (x_{n+1} - x^*) \leq 2LD$ we see

$$\mathbb{E}|a \cdot (x_{n+1} - x^*)| \leq 2LD \cdot P(x_{n+1} \notin U) + \int_{x_{n+1} \in U} |a \cdot (x_{n+1} - x^*)| dP.$$

Lemma 6 says the integral is at most

$$\int_{x_{n+1} \in U} \frac{4\|a\|}{m} \|N(x_{n+1}) - N(x^*)\|^2 dP \leq \frac{4\|a\|}{m} \mathbb{E}|N(x_{n+1}) - N(x^*)\|^2.$$

and we see $\mathbb{E}|a \cdot (x_{n+1} - x^*)|$ is at most

$$2LD \cdot P(x_{n+1} \notin U) + \frac{4\|a\|}{m} \mathbb{E}\|N(x_{n+1}) - N(x^*)\|^2.$$

Lemma 8 says the series $\sum_{i=1}^{\infty} P(x_i \notin U)$ converges. Lemma 7 gives constants $k \geq 0$ and $K = \frac{36}{\|a\|^2}\left(\frac{\sqrt{8\pi}DR}{\eta} + 4R^2\right)$ such that $\mathbb{E}\|N(x_{n+1}) - N(x^*)\|^2 \leq K/n + 8e^{-kn}$. Hence we get

$$\mathbb{E}\left[\sum_{i>n_0}^{N} a \cdot (x_i - x^*)\right] = O(1) + \frac{4\|a\|}{m} \sum_{i=n_0}^{N} (K/i + 8e^{-ki}) = O(1) + \frac{4\|a\|}{m} \int_{n_0}^{N} (K/x + 8e^{-kx}) dx$$

$$= O(1) + \frac{4\|a\|}{m} K \log N = O(1) + \frac{4\|a\|}{m} K \log N = \frac{144}{m\|a\|}\left(\frac{\sqrt{8\pi}DR}{\eta} + 4R^2\right) + O(1).$$

$\square$

# 4  Equivalence of $\mathbb{E}[R_N]$ and $\mathbb{P}[R_N]$

Our Theorem 1 gives an $O(\log N)$ bound for pseudo-regret. This seems weaker than the $O(\log N)$ expected regret bound of [20]. Here we show the two types of order bounds are in fact equivalent for strongly convex domains.

**Theorem 2.** Under Assumptions 1 and 2, Algorithm 1 with parameter $\eta > 0$ gives

$$\mathbb{E}[R_N] - \mathbb{P}[R_N] \leq \frac{31R^2}{m\|a\|} = O\left(\frac{L^2}{m\|a\|}\right).$$

Theorem 2 holds even for non smooth domains. We need only the following result of Vial [35].

**Lemma 9.** (Vial Theorem 1(v)) Suppose $Z$ is $m$-strongly convex. For all $x, y \in \partial Z$ and outward unit normals $p, q$ at $x, y$ respectively we have $\|x - y\| \leq \frac{1}{m}\|p - q\|$.

To prove Theorem 2 we first put the left-hand-side in a convenient form.

**Lemma 10.** Let $y^* \in \operatorname{argmin}\big\{\sum_{i=1}^{N} a_i \cdot x : x \in X\big\}$. Under Assumption 2 we have

$$\mathbb{E}[R_N] - \mathbb{P}[R_N] \leq \sqrt{\mathbb{E}\|x^* - y^*\|^2}\sqrt{\mathbb{E}\big\|\sum_{i=1}^{N}(a_i - a)\big\|^2}.$$

*Proof.* By definition $\mathbb{E}[R_N] - \mathbb{P}[R_N]$ equals $\mathbb{E}\big[\sum_{i=1}^{N} a_i \cdot (x_i - y^*)\big] - \mathbb{E}\big[\sum_{i=1}^{N} a \cdot (x_i - x^*)\big] = \mathbb{E}\big[\sum_{i=1}^{N}(a_i - a) \cdot x_i\big] + \mathbb{E}\big[\sum_{i=1}^{N} a_i \cdot (x^* - y^*)\big]$. To see the first term is zero recall each $x_i$ is a function of $a_1, \ldots, a_{i-1}$. Since $a_1, a_2, \ldots$ are independent we see each $x_i$ is independent of $a_i$ and independent of $a_i - a$. Thus the expectation distributes over the product. Since $\mathbb{E}[a_i - a] = 0$ the first term vanishes. Write the second term as $\mathbb{E}\big[\sum_{i=1}^{N}(a_i - a) \cdot (x^* - y^*)\big] - \mathbb{E}\big[\sum_{i=1}^{N} a \cdot (y^* - x^*)\big]$. Since $x^*$ minimises $a \cdot x$ the second term is positive and we get $\mathbb{E}[R_N] - \mathbb{P}[R_N] \leq \mathbb{E}\big[\sum_{i=1}^{N}(a_i - a) \cdot (x^* - y^*)\big]$. To complete the proof apply Cauchy-Schwarz to the $L^2$ norm. $\square$

---

[4]According to the adversarial bound the regret before turn $n_0$ is $O(\sqrt{n_0}) \leq O(D/\eta\|a\|)$.

To prove Theorem 2 we need the following three results from Appendix III. Like in Section 3 the first two follow from the vector martingale theorems of Pinelis [28]. The third result is well known but we were unable to find a sufficiently general proof in the literature.

**Proposition 2.** Suppose $X_1, X_2, \ldots \in \mathbb{R}^d$ are independent random variables with each $\mathbb{E}[X_n] = 0$ and $\|X_n\| \leq R$. For each $t > 0$ we have

$$P\left(\left\|\frac{1}{n}\sum_{i=1}^n X_i\right\| > t\right) \leq 2\exp\left(-\frac{t^2}{2R^2}n\right).$$

**Lemma 11.** For $y^* = \arg\min\left\{\sum_{i=1}^N a_i \cdot x : x \in X\right\}$ the Online Lazy Gradient Descent actions give

$$\mathbb{E}\|N(x^*) - N(y^*)\|^2 \leq \frac{144R^2}{\|a\|^2 N} + 4\exp\left(-\frac{\|a\|^2}{8R^2}N\right).$$

**Lemma 12.** For $X$ a nonnegative random variable we have $\mathbb{E}[X] = \int_0^\infty P(X > x)dx$.

*Proof of Theorem 2.* Proposition 2 gives $P\left(\|\sum_{i=1}^N (a_i - a)\| > t\right) \leq 2\exp\left(-\frac{t^2}{2NR^2}\right)$ for all $t \geq 0$. Replace $t$ with $t^2$ to see $P\left(\|\sum_{i=1}^N (a_i - a)\|^2 > t\right) \leq 2\exp\left(-\frac{t}{2NR^2}\right)$. Lemma 12 says $\mathbb{E}\|\sum_{i=1}^N (a_i - a)\|^2 \leq \int_0^\infty P(\|\sum_{i=1}^N (a_i - a)\|^2 > t)dt \leq 4NR^2$. Hence Lemma 10 gives

$$\mathbb{E}[R_N] - \mathbb{P}[R_N] \leq 2R\sqrt{N}\sqrt{\mathbb{E}\|x^* - y^*\|^2}.$$

For the second factor Lemma 9 gives $\sqrt{\mathbb{E}\|x^* - y^*\|^2} \leq \frac{1}{m}\sqrt{\mathbb{E}\|N(x^*) - N(y^*)\|^2}$. Lemma 11 gives $\mathbb{E}\|N(x^*) - N(y^*)\|^2 \leq K/N + 4e^{-kN}$ for the constants $K = 144R^2/\|a\|^2$ and $k = \|a\|^2/8R^2$. Hence $\sqrt{\mathbb{E}\|x^* - y^*\|^2} \leq \frac{1}{m}\sqrt{K/N + 4e^{-kN}} \leq \frac{1}{m}\sqrt{\frac{K}{N}} + \frac{2}{m}e^{-(k/2)N}$ and so

$$\mathbb{E}[R_N] - \mathbb{P}[R_N] \leq \frac{2R\sqrt{K}}{m} + \frac{4R}{m}\sqrt{N}e^{-(k/2)N} = \frac{24R^2}{m\|a\|} + \frac{4R}{m}\sqrt{N}e^{-(k/2)N}. \tag{3}$$

To bound the last term consider the function $\sqrt{x}e^{-\beta x}$ for $\beta \geq 0$. It is straightforward to see the function is increasing and then decreasing. By differentiating we see the maximiser is $x = 1/2\beta$ and the function is at most $1/\sqrt{2e\beta}$. For $\beta = k/2$ we see $\sqrt{N}e^{-(k/2)N} \leq 1/\sqrt{ke} = 2\sqrt{2}R/\sqrt{e}\|a\|$. Thus the right-hand-side of (3) is at most $\left(24 + \frac{8\sqrt{2}}{\sqrt{e}}\right)\frac{R^2}{m\|a\|} \leq \frac{31R^2}{m\|a\|}$. □

Finally we can combine Theorems 1 and 2:

**Theorem 3.** Under Assumptions 1 and 2, Algorithm 1 with parameter $\eta > 0$ gives

$$\mathbb{E}[R_N] \leq \frac{144}{m\|a\|}\left(\frac{\sqrt{8\pi}DR}{\eta} + 4R^2\right)\log N + O(1).$$

## 5 Numerical Simulations

Here we present numerical simulations for Online Lazy Gradient Descent, using a range of strongly convex domains and i.i.d opponents. The higher dimensional simulations ran on the order of minutes, due to use of an all-purpose Python package to compute minimisers. Performance time can be improved by using the Franke Wolfe method of [17] to simplify the minimisation problems.

$\ell_p$**-Norm Balls.** Here we examine performance for domains $X = \left\{x \in \mathbb{R}^d : \left(\sum_{i=1}^d |x(i)|^p\right)^{1/p} \leq 1\right\}$. For $p \in (1, 2]$ Corollary 1 of [17] says $X$ is $(p-1)d^{\frac{1}{2} - \frac{1}{p}}$ strongly convex.

For simplicity we use stepsize $\eta = 1$. When searching for worst-case performance it is enough, by symmetry of the domain, to only consider $\mathbb{E}[a_n] = a$ with nonnegative nondecreasing components. We consider $a = b/\|b\|$ for $b(i) = (i/d)^r$ and a range of parameters $r \geq 0$. For example three degenerate cases are $a = \frac{1}{\sqrt{d}}(1, \ldots, 1)$ for $r = 0$; $a = \sqrt{\frac{2}{d(d+1)}}(1, 2, \ldots, d)$ for $r = 1$ and $a = (0, \ldots, 0, 1)$ for $r \to \infty$. We use cost vectors $a_n = a + \mu_n$ for two types of noise:

(1) $\mu_n(j) = \frac{1}{\sqrt{d}}(B_1^n, \ldots, B_d^n)$          (2) $\mu_n(j) = (0, \ldots, 0, B_d^n)$.

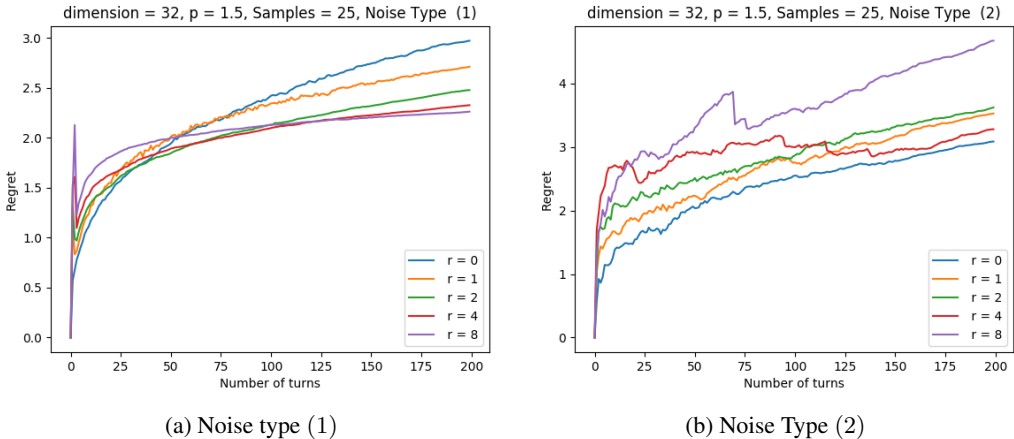

(a) Noise type $(1)$                              (b) Noise Type $(2)$

Figure 1: Online Lazy Gradient Descent on the $\ell_{1.5}$ unit ball.

Here $B_j^n$ are independent and take values $\pm 1$ with equal probability. In both cases $\|\mu_n\| = 1$. Noise type $(1)$ perturbs all components equally and individual perturbations are small for large $d$. Type $(2)$ perturbs only the largest with size 1. We run several instances (samples) of each online problem and plot the average pseudo-regret over instances.

Figure 1(a) shows that for noise type $(1)$ performance degrades as $r$ shrinks. This is consistent with the picture for $d = 2$ and $r = 0$ where the minimiser $(2^{-1/p}, 2^{-1/p})$ also minimises the curvature, and how Theorem 1 gives a worse bound for small curvature. For noise type $(2)$ Figure 1(b) shows larger $r$ gives worse performance. This can be attributed to how $r$ is approximately $(0, \dots, 0, 1)$ and the noise is focused on the final component rather than spreading over all components evenly.

For smaller of $d, p$ the pattern is the same but takes longer to emerge. In Figure 3, Appendix I, we consider the worst cases in Figure 1 and vary the dimension. For $r = 0$ and noise type $(1)$ the performance improves slightly with large $d$. This is perhaps due to the higher dimension having a regularising effect on the noise. For $r = 8$ and noise type $(2)$ the performance deteriorates with large $d$. This is expected since the curvature $(p-1)d^{\frac{1}{2} - \frac{1}{p}}$ shrinks in high dimensions. In Figure 4, Appendix I, we vary the parameter $p$. In both worst cases the performance deteriorates as $p \to 1$ though it also takes longer for the worse performance to emerge.

For all cases the growth rates are orders of magnitude below those suggested by Theorem 1. For example in the setup of Figure 1 we have $p = 1.5$, $d = 32$ and the convexity parameter is $m = (p-1)d^{\frac{1}{2} - \frac{1}{p}} = 0.5 \cdot 32^{-\frac{1}{4}} \simeq 0.21$. This makes the coefficient in Theorem 1 approximately 6180.

**Comparison with Greedy Gradient Descent.** Theorem 3 says Online Lazy Gradient Descent gives logarithmic expected regret on a strongly convex domain against i.i.d cost vectors. One can ask does Online Greedy Gradient Descent with update $x_{n+1} = \Pi_X \left( x_n - \eta a_n / \sqrt{n} \right)$ give the same bound. Figure 2(b) suggests otherwise. The worst-case regret of Greedy Gradient Descent seems to be $\Omega(\sqrt{n})$. The coefficients $D_1, C_1$ in Figure 2(b) are chosen to match the regret on turns 1000 and 500, and the $\Theta(\sqrt{n})$ fit is very close over the entire range. The coefficient $C_2$ is chosen to match on turns 0 and 1000 and is a worse fit.

**Schatten Matrix Norms.** Schatten matrix norms are well-studied as regularisers in the context of machine learning. For example see [3, 21] and the citing works.

**Definition 2.** Given a real $n \times m$ matrix $A$ there is a unique decomposition $A = U\Sigma V$ such that $V, U$ are real orthogonal, $U$ is $n \times n$, $V$ is $m \times m$, and $\Sigma$ is $n \times m$ with real diagonal entries $\sigma_1 \geq \sigma_2 \geq \dots \geq \sigma_{\min(n,m)}$ and all other entries zero. The $\sigma_i$ are called the singular values of $A$. For any $p \geq 0$ define the Schatten $p$-norm $\|A\|_{S(p)} = \|\sigma\|_p$ as the $p$-norm of the vector of singular values.

For simplicity we only consider square matrices. Corollary 2 of [17] says the Schatten unit balls $B_p(1) = \{X \in \mathbb{R}^{d \times d} : \|X\|_{S(p)} \leq 1\}$ are $(p-1)d^{\frac{1}{2} - \frac{1}{p}}$ strongly convex. When searching for worst case performance the following lemma says it is enough to only consider when $\mathbb{E}[a_n]$ is a diagonal matrix. See Appendix I for proof.

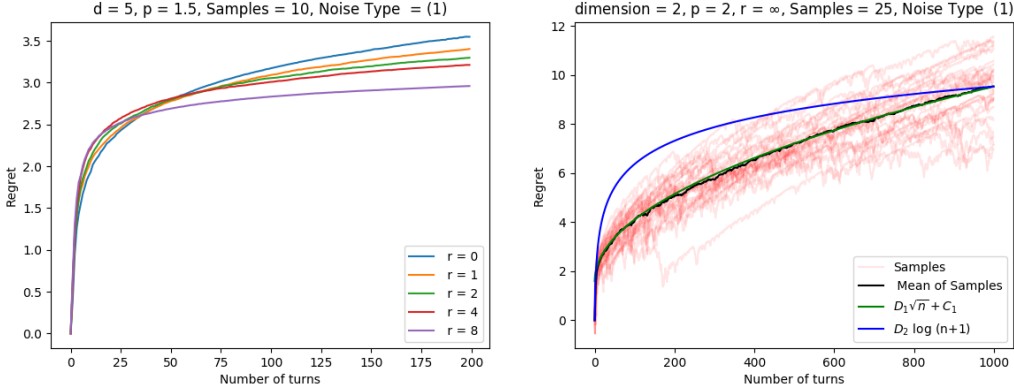

(a) Lazy Gradient Descent on the Schatten unit ball    (b) Greedy Gradient Descent on the Euclidean unit ball

Figure 2

**Lemma 13.** Suppose $a_1, a_2, \ldots \in \mathbb{R}^{d \times d}$ are i.i.d cost vectors with $\mathbb{E}[a_n] = a$. There is another set of i.i.d cost vectors $b_1, b_2, \ldots$ such that $\mathbb{E}[b_n] = b$ is diagonal and $\|a_n - a\| = \|b_n - b\|$; $\|a_n\| = \|b_n\|$; $\|a\| = \|b\|$ and the expected regret of playing Online Lazy Gradient Descent (Algorithm 1) on $B_p(1)$ against $a_1, a_2, \ldots$ with $x_0 = 0$ is the same as playing against $b_1, b_2, \ldots$ with $x_0 = 0$.

Likewise by symmetry we can assume the diagonal entries are nonnegative and nondecreasing. For simplicity we use stepsize $\eta = 1$. We consider $a = b/\|b\|$ for $b = \text{diag}\big((1/d)^r, (2/d)^r, \ldots, 1\big)$ and a range of parameters $r \geq 0$. We consider cost vectors $a_n = a + \mu^n$ and noise $\mu^n = \nu^n/\|\nu^n\|$ for three types of $\nu^n$. Here $B_{ij}^n$ are independent and take value $\pm 1$ with equal probability.

(1) $\nu_{ij}^n = B_{ij}^n$ for all $i, j$

(2) $\nu_{ij}^n = B_{ij}^n$ for all $i$ and some uniformly chosen $j$; and $\nu_{ik}^n = 0$ for all $k \neq j$.

(3) $\nu_{ij}^n = B_{ij}^n$ for some uniformly chosen $i, j$; and $\nu_{lk}^n = 0$ for all $(l, k) \neq (i, j)$.

Compared to the examples for $\ell_p$ balls, fewer samples are needed to obtain smooth curves. For noise type (1) Figure 2(a) shows the worst performance occurs for $r = 0$. The different types of noise make minimal difference to performance. See Figure 5, Appendix I. Thus when varying $p$ and $d$ we only consider noise type (1). In Figure 6, Appendix I, we see large $d$ leads to slightly better performance, and small $p$ leads to worse performance. Though the infuence of $p$ and $d$ is minimal compared to that in Figures 5 and 2 for $\ell_p$-balls.

For Schatten balls diagonal noise matrices seem to be the most difficult to deal with. In that case all $a_n$ are also diagonal and the Schatten norm reduces to the $p$-norm of the diagonal entries.

## 6   Conclusion

Online Lazy Gradient descent on a strongly convex domain is universal in the sense that it achieves $O(\log N)$ regret against i.i.d opponents. This is in addition to the well-known $O(\sqrt{N})$ regret bound against adversarial opponents. Moreover for strongly convex domains, order bounds for pseudo-regret and expected regret are equivalent. For various types of noise on $\ell_p$ and Schatten unit balls, the growth coefficients are orders of magnitude smaller than those suggested by our analysis.

**Disclosure of Funding.** This work was supported by Science Foundation Ireland grant 16/IA/4610.

**Broader Impact.** This is theoretical work on special cases of online optimisation, with no particular application in mind. We do not forsee the work having any direct impact on society. The only ethical concerns are those related to the use of online optimisation in general.

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
