# OpenReview forum: "The Lazy Online Subgradient Algorithm is Universal on Strongly Convex Domains"
_NeurIPS.cc/2021/Conference — NeurIPS 2021 Poster_

### Official Review · Reviewer_grJg · 2021-07-14

**Rating:** 6
**Confidence:** 4

**Summary:**

They authors show that for linear optimization on strongly convex domains the lazy subgradient method achieves a best-of-both-worlds behavior of a expected regret bound of O(\sqrt{N}) and O(\log(N)) for adversarial and stochastic adversaries, respectively. The best known prior bounds were O(\sqrt{log(N)}) adversarial regret with a more complicated algorithm (based on A,B-prod). On a technical side, the analysis for lazy subgradient is much more intricate and considers loss contributions in the direction of the comparator separately from other directions, and may be of interest in other online linear optimization settings.

Simulations are provided that convincingly argue for the proposed method in the proposed setting.


**Limitations And Societal Impact:**

Yes.

**Main Review:**

The introduction is generally well written and the contributions are clearly explained and well-positioned in the literature. I think the results are impressive and that the ideas borrowed from differential geometry would be a great introduction to the neurips community.

However, I found the sections requiring differential geometry to be hard to understand. For example, what is y on line 117 (a assume it is a typo for \alpha)? Why is the statement on l120 "the line from y_n...to any x has slope of absolute value Omega(\sqrt{n}) true (in particular, please indicate that this is a high probability statement)? Why does alpha_n = beta_n (this only holds when sum a_n / n is not in \mathcal X, which does happen with high probability), etc. A picture would help, too. You clearly have a strong geometric intuition, but it does not come through clearly.

While I'm excited to see ideas from higher branches of mathematics percolate into this conference, their rarity means that there is significant burden on the authors to explain the ideas in way comprehensible to the community. I'm afraid that I don't feel this burden has been met in this submission. I feel that even the level of detail provided in the appendix may not be sufficient, but the main body definitely needs to be fleshed out. Some other questions I (and I would imagine other readers) would have are:
-Since \Vert\cdot\Vert is defined as the Euclidean norm, is geodesic distance in Definition 1 is always the Euclidean distance?
- Please explain l130: the normal direction to the surface. Which surface?
- How do you define a coordinate patch and what are its derivatives?
- Additionally, most of the NeurIPS community will not think of derivatives as linear maps, so the notation \nabla N(z)v will probably be confusing.

typos/suggestions:
you didn't define before using it

It would also be good to delineate which lemmas and propositions are novel and which are standard results from differential geometry.

So in summary, I'm very positive about the paper this submission could be if a better introduction to differential geometry were provided.

Post-rebuttal: I believe the authors will revise the paper enough to clarify the presentation, so I increased my score.

**Time Spent Reviewing:**

3

---

> ### Author Response · Authors · 2021-08-30
> **Response to Reviewer Concerns (previously mistakenly submitted under Review Feedback)**
>
> Thank you for your feedback and praise. We will attempt to improve the presentation according to your comments. Though of course it is a nontrivial task to both introduce and apply these techniques -- which we believe are unfamiliar to the readership at large -- while also keeping to the page limit in the main paper.
>
> "However, I found the sections requiring differential geometry to be hard to understand."
>
> This is valuable feedback, since this section is trying to give an intuitive and geometric overview of our main argument.
>
> "For example, what is y on line 117 (a assume it is a typo for \alpha)?"
>
> This is a typo. y < 1 should be \beta < 1 , thus we only look at the part of the epigraph below the line y = 1.
>
> "Why is the statement on l120 "the line from y_n...to any x has slope of absolute value Omega(\sqrt{n}) true"
>
> It is because the vector \sum a_i has length increasing linearly, but has x-component \sum (a_i - a) increasing at O(\sqrt n) rate. Hence the y-component increases at least linearly, and the slope has order Omega( n / \sqrt n) = Omega(\sqrt n). Perhaps this terminology would be easier to understand in the main paper.
>
> "(in particular, please indicate that this is a high probability statement)?".
>
> Yes, the conclusion only holds whp, as do all further statements in this proof.
>
> "Why does alpha_n = beta_n (this only holds when sum a_n / n is not in \mathcal X, which does happen with high probability)"
>
> Because \sum a_i grows linearly hence \sum a_i / \sqrt n grows like Omega(\sqrt). Hence \sum a_i / \sqrt n is eventually outside the domain and projects onto the boundary. Since \sum a_i is below the x-axis it projects onto the graph y = x^2.
>
> "While I'm excited to see ideas from higher branches of mathematics percolate into this conference, their rarity means that there is significant burden on the authors. . ."
>
> We agree the burden is on the authors to make the new material accessible.
>
> "Some other questions I (and I would imagine other readers) would have are: -Since \Vert\cdot\Vert is defined as the Euclidean norm, is geodesic distance in Definition 1 is always the Euclidean distance?"
>
> In this paper the geodesic distance is always wrt the Euclidean norm. In a broader context, for embedded manifolds, it usually refers to the Euclidean distance. For intrinsically defined manifolds -- such as those in general relativity or abstract differential geometry -- the geodesic distance is defined using a more general "metric tensor" that measures infinitesimal distances and depends on the manifold in question. There is also a broader theory of geodesic spaces defined with more general notion of distance, but this is outside the purview of differential topology.
>
> "Please explain l130: the normal direction to the surface. Which surface?"
>
> The boundary surface to the domain.
>
> "How do you define a coordinate patch and what are its derivatives?"
>
> Our definition of a coordinate patch is contained in Assumption 1. It is a representation of a region of the surface as the zero set of some function. The derivative of a coordinate patch is just the derivative of that function.
>
> Note this terminology is different to the standard literature, where a "coordinate patch" refers to a function from the region to some subset of R^{d-1}. This is an abuse of notation, however we don't see a major issue, since it is an introductory result in differential geometry that the existence of the two types of "coordinate patches" are equivalent.
>
> "Additionally, most of the NeurIPS community will not think of derivatives as linear maps, so the notation \nabla N(z)v will probably be confusing."
>
> At this point in the paper \Nabla(z) is just the matrix of partial derivatives and N(z)v is applying the matrix to a vector.. i.e N is a vector valued function; \Nabla(z) is the matrix of partial derivatives, and N(z)v is applying the matrix of partial derivatives to the vector v.
>
> "It would also be good to delineate which lemmas and propositions are novel and which are standard results from differential geometry."
>
> We go so far as to claim that all of the differential geometry arguments in the appendix are routine. This is discussed after line 135.
>
> The reason we include such a lengthy appendix, is that most of the literature is about intrinsically defined manifolds, and so cannot be cited directly here.
>
> Instead we attempt to re-prove more concrete results that can be more easily digested by the readership. For example in the intrinsic context the analogue of the constant M in Lemma 4 is the largest eigenvalue of the curvature operator, i.e rate of change of the normal vector as we move along the surface. To avoid discussing the operator in the main paper, we instead use the larger costant M that also bounds the change as we move perpendicular to the surface.
>
> We believe Lemmas 1 and 6 are new. Lemma 1 bridges the gap between the differential geometry results, and our setting. Lemma 6 introduces the expected cost vector.

---

### Official Review · Reviewer_37jk · 2021-07-16

**Rating:** 8
**Confidence:** 2

**Summary:**

The authors study online linear optimization and show that the Lazy Online Subgradient Algorithm is *universal* on strongly convex domains: it simultaneously achieves the optimal $O(\sqrt T)$ regret when the costs are adversarial and the optimal $O(\log T)$ regret when the costs are stochastic. This result is a clear improvement over previous work of Huang et al, who described a more complex algorithm and didn't achieve the optimal rates. The proof is based on an intricate analysis, drawing from differential geometry, of the points selected by the algorithm, and in particular, how much of each step affects regret against the minimizer $x^*$ and how much is orthogonal to the direction $x_i - x^*$ and doesn't affect regret. This technique may be of interest to the broader ML community.

**Ethical Concerns:**

No problem here.

**Limitations And Societal Impact:**

No problem here.

**Main Review:**

This is a neat paper, and is especially cool because it uses differential geometry in a nontrivial way to say something non-obvious about the regret when the costs are stochastic. It is well known that OGD-style algorithms achieve $O(\sqrt T)$ regret in the worst-case, and FTL gives low regret when the costs are stochastic, but this paper shows that in fact gradient-based algorithms can also achieve low regret when the costs are stochastic, at least if the domain is sufficiently smooth.

A few questions/suggestions: first, I am interpreting the lazy online subgradient algorithm as special case of OGD, but with subgradients instead of gradients. In this case, the costs are linear, so one can just use gradients without resorting to subgradients. Is that correct? How does the lazy version differ from the greedy version? I think it would be helpful to add a few sentences in the introduction explaining the similarities/differences between the lazy online subgradient algorithm, the greedy online subgradient algorithm, and OGD.

I believe that the bandit result of Zimmert and Seldin (2018) only works if one sets the step size knowing whether the instance is adversarial or stochastic. The authors may wish to point out that their analysis does not suffer from this restriction; the same step-size gets the optimal rate in both settings.

It may be interesting to say something about what goes wrong when the domain is not strongly convex. Can you show an example of a domain where the regret is $\Omega(\log T)$?

Trivial points: the standard notation in online learning is to use $\Pi_S(x)$ to denote the projection of $x$ onto the set $S$. The authors use $P_S(x)$ and may wish to consider altering their notation in the final version. Similarly, people generally use "orthogonal" instead of "perpindicular", especially when talking about geometries beyond $\ell_2$.

Verdict: Surprising result, clean analysis, timely problem. Clear accept.

**Time Spent Reviewing:**

1

---

> ### Author Response · Authors · 2021-08-30
> **Response to Reviewer Concerns (previously mistakenly submitted under Review Feedback)**
>
> Thank you for your feedback and praise of the paper. We also think the result is pretty neat in its simplicity.
>
> "A few questions/suggestions: first, I am interpreting the lazy online subgradient algorithm as special case of OGD,"
>
> I think typically OGD refers to what we call the greedy (sub)gradient algorithm. In that algorithm we move downwards in the direction of the (sub)gradient at the current action, and then project back onto the domain to get the next action. The lazy algorithm is conceptally similar, with the key difference that we sum all the past (sub)gradients before projecting.
>
> "In this case, the costs are linear, so one can just use gradients without resorting to subgradients. Is that correct?"
>
> Agreed the terminology "Lazy Subgradient" stands to be improved. Why call it Lazy Subgradient when we in fact have gradients? Perhaps a better title is "Lazy Online Gradient Descent is Universal on Strongly Convex Domains" and then change "subgradient" to "gradient" throughout.
>
> "I believe that the bandit result of Zimmert and Seldin (2018) only works if one sets the step size knowing whether the instance is adversarial or stochastic. The authors may wish to point out that their analysis does not suffer from this restriction; the same step-size gets the optimal rate in both settings."
>
> Are you referring to "Tsallis-INF: An Optimal Algorithm for Stochastic and Adversarial Bandits"? From what I understand, the 2021 jmlr version of the paper has discussion after Theorem 1 stating that we don't need to know the setting in advance. Though Theorems 3 and 4 discuss versions of their algorithm with parameter alpha =! 1/2 and require setting the stepsize based on knowledge of the setting.
>
> "It may be interesting to say something about what goes wrong when the domain is not strongly convex. . . "
>
> Space permitting, this is worth mentioning, since at the other end of the spectrum, we also get O(1) regret in the setting where the domain is a polytope. The simplest examplesthe two extremes in between are the epigraphs of functiona x^a for suitably chosen a > 0. Choosing a appropriately we can get regret as bad as \Omega( N^ {1/2 - \epsilon}) for any positive epsilon.
>
> "Trivial points. . ."
>
> Agreed. We will change the notation in the new version. Though in this case we are talking about Euclidean Geometry.

---

### Official Review · Reviewer_THEd · 2021-07-16

**Rating:** 6
**Confidence:** 4

**Summary:**

The paper considers the problem of online linear optimization under a full information feedback setting on a strongly convex domain. In this setting, the authors show that the classical lazy subgradient method is optimal both in the adversarial and stochastic settings.  In particular, the authors show that for the same choice of hyper-parameters, the lazy sub-gradient method achieves O(\sqrt{N}) regret in the adversarial setting and O(log{N}) regret in the stochastic setting. To show this result, the authors have to do a more nuanced analysis of the algorithm (that relies on differential geometry methods) than existing works.



**Limitations And Societal Impact:**

the authors adequately addressed the potential negative societal impact of their work

**Main Review:**

Although the problem setting is very narrow, I feel the result is interesting. Unlike past works which come up with complex algorithms, the paper shows that a simple lazy sub-gradient method works well both in adversarial and stochastic settings.

I found the paper to be a little bit terse. In particular, I'd appreciate a little bit more insights into proof techniques (maybe a high-level proof sketch before jumping into the proof of Theorem 1).  Also, I think a more thorough literature review needs to be performed (more on this below).

Here are my main comments on the paper:
1)  Regarding the experiments on the lazy subgradient method presented in Figures 1,2: similar to Figure 2b, can the authors try to fit a curve through the points (fit both D1*sqrt{n}+C1 and D2*log(n+1)+C2) and see which is a better fit? Do the experiments support the theory that the regret scales with log{N} in the stochastic setting?

2) Regarding the greedy subgradient method: in line 244, the authors mention that the worst-case regret of this technique could be Omega(sqrt{N}) in the stochastic setting. This seems a bit counterintuitive to me. The greedy subgradient method is nothing but the projected stochastic gradient descent (SGD) method.  Projected SGD is a well-studied problem. Many works have shown that it converges at O(1/N) rate to an optimum for strongly convex losses (for example, see the works of Francis Bach). Given this, I find it a little bit hard to believe that for linear losses on strongly convex domains, SGD only converges at O(1/sqrt{N}) rate to an optimum. Can the authors explain this?

3) I believe a more thorough literature review needs to be performed. One line of work that is missed is the optimistic online learning algorithms of [1]. Look at section 4 in [1] where the algorithms learn to adapt to the adversary. Can these algorithms not be made to adjust to the stochastic setting and achieve O(log{N}) regret?
Given the relation between lazy subgradient method and SGD, some relevant literature on SGD should also be cited.

4) As mentioned before, the setting considered in this paper is a bit narrow. So it would be good to see if the results can be generalized. Can the results be generalized to general convex functions on strongly convex domains? Will the same results hold for lazy Online Mirror Descent?

Finally, please avoid revealing the funding sources in the future. This defeats the purpose of the double-blind review process.


[1] Rakhlin, A. and Sridharan, K., 2013, June. Online learning with predictable sequences. In Conference on Learning Theory (pp. 993-1019). PMLR.

**Time Spent Reviewing:**

8

---

> ### Author Response · Authors · 2021-08-30
> **Response to Reviewer Concerns (previously mistakenly submitted under Review Feedback)**
>
> Thank you for your feedback. We will respond to your comments in order.
>
> We have attempted to give a high-level overview of our main proof at the bottom of page 2. We will attempt to connect this overview more explicitly to the sequence of lemmas used to prove Theorem 1.
>
> 1. This kind of curve fitting is surprisingly difficult. It seems neither of the curves of the form C1 \sqrt n + D2 or C2 log n + D2 fit the data in Figure 1 well. Even using more complex fitting methods, such as least-squares over the whole domain, neither of the two types of curves matches well on the entire domain.
>
> 2. If I understand correctly, you are referring to the result (Adaptivity of Averaged Stochastic Gradient Descent to Local Strong Convexity for Logistic Regression) for example that sgd (with O(1/\sqrt n) stepsize) against the function x^2 on the interval [-1,1] should give squared gradients decaying to zero at O(1/n) rate; and since the squared gradients are the same order as the regret this should give O(log N) regret.
>
> This problem has in common with our Greedy problem on the Euclidean sphere, that both setups have the squared horizontal distance from the origin as the regret. However on closer inspection the updates don't seem to be equivalent. The sgd update on the interval takes the gradient at the current point plus the noise term, rescales by the stepsize and moves horizontally that far from the previous point. The greedy update on the sphere takes only the noise term, and since we are projecting, the horizontal movement depends on the previous point.
>
> Granted this doesn't give any understanding to why Greedy gives such bad performance on the sphere. This surprises us as well, and we don't have a nice answer for why it happens. The recursive nature of the greedy algorithm seems more resistant to the sort of proof in this paper, where we focus on the action rather than the regret. Proofs that focus on the regret rather than the action are counter-intuitive to begin with.
>
> 3. We are familiar with such optimistic learning algorithms. They can be attempted here, using for example the prediction b_{n+1} = (a_1+ . . . + a_n)/n for a_{n+1} and adaptive stepsize \sqrt{ \sum | b_n - a_n|^2 }. The standard optimistic bounds are then of order \sqrt{ \sum | b_n - a_n|^2 }. The problem is the radicand contains a_n in the n-th term rather than a. If we had a-terms then each term would be order 1/n and get log(N) bounds. Since we have a_n instead, what happens is the estimate b_n approaches a for large n, and the bound has order only \sqrt{ \sum | a - a_n|^2 } which looks something like (STANDARD DEVIATION)* \sqrt N.
>
> 4. Agreed the setting is quite narrow. It is not obvious whether our methods generalise to, for example, noisy access to the gradients of a fixed convex objective. The missing step would be to show how the gradients at the points x_1,x_2, . . . approximate the gradient of the objective function (in the larger space) at the minimiser. This is obviously not true for an arbitrary sequence x_1,x_2, . . . so we would need some special feature of how the actions are generated.
>
> It sounds even harder to generalise to mirror descent with non-Euclidean norms, since the differential geometry used here makes use of inner products on the tangent space, which generates the euclidean norm. For example the p-norms for cannot arise as such (unless p = 2).

---

### Official Review · Reviewer_e3cw · 2021-08-03

**Rating:** 4
**Confidence:** 2

**Summary:**

The authors study online linear optimization in the case of known strongly convex constraints sets. They show that the simple method of online subgradient descent, which is known to achieve $O(\sqrt{N})$ regret in the adversarial case, also achieves $O(\log(N))$ regret in the i.i.d. case. This result is more elegant than and slightly improves upon existing results in the strongly convex case.

**Main Review:**

Strengths:
- The paper improves upon the state of the art results for online linear optimization in the strongly convex case
- The result is elegant as it pertains to the basic online subgradient descent method instead of a more complicated algorithm
- Online linear optimization is a significant problem

Weaknesses:
- My main concern is about the significance of the results given that the paper entirely focuses on the specific case of strongly convex domains. I'm not sure that the authors do enough to justify a paper that focuses entirely on this special case, and therefore I find the significance to be largely limited
- Additionally, although it is nice that the results pertain to a simpler and more popular algorithm than prior results, the complexity bounds only improve slightly by a factor of $\sqrt{log{N}}$ in the adversarial case.
- The writing of the paper can be improved substantially, both for clarity and language/grammar etc.

**Time Spent Reviewing:**

2 hours

---

> ### Author Response · Authors · 2021-08-30
> **Response to Reviewer Concerns (previously mistakenly submitted under Review Feedback)**
>
> Thank your feedback on the paper. We agree the strongly-convex setting is quite restrictive, and we are not aware of many practical problems where it is natural to consider a strongly convex domain.
>
> What we believe is interesting and worthwhile, however, is that our main result shows extra properties of Lazy subgradient that we get "for free". i.e even if the extra conditions are restrictive, we do not need to modify the algorithm or know if the extra conditions hold in advance, in order to benefit from the improved performance.
>
> It is also interesting that the improved performance is not shared by the Greedy subgradient algorithm. This is a reason to favour Lazy over Greedy -- which conflicts with our experience of the literature, where it seems Greedy-type algorithms are more studied than Lazy-type. We are not aware of a theoretical justification for this focus.

---

### Decision · Program_Chairs · 2021-09-27

**Decision:**

Accept (Poster)

**Comment:**

The paper has received mixed reviews; while some reviewers found the results not super significant (essentially simplifying existing results in the literature) and the problem setting restrictive, others found the realization that simple plain lazy OGD is “universal” compelling, and the proof interesting and overall elegant.  All considered---and while I agree with the criticism---I think that there is a fair chance that the paper’s observations will draw some attention and promote further investigation, and I support its acceptance.

One remaining concern the authors should carefully consider is that the differential geometry background, and more importantly - the motivation and justification for appealing to such techniques, is lacking.  I trust the authors to improve this part of their exposition for their final version.